# Rapid Intensification of Typhoon Hato (2017) over Shallow Water

**Iam-Fei Pun** [1,*], **Johnny C. L. Chan** [2], **I.-I. Lin** [3], **Kelvin T. F. Chan** [4,5], **James F. Price** [6], **Dong Shan Ko** [7], **Chun-Chi Lien** [3], **Yu-Lun Wu** [3] **and Hsiao-Ching Huang** [3]

1. Graduate Institute of Hydrological and Oceanic Sciences, National Central University, Taoyuan 32001, Taiwan
2. Guy Carpenter Asia-Pacific Climate Impact Centre, School of Energy and Environment, City University of Hong Kong, Hong Kong, Hong Kong
3. Department of Atmospheric Sciences, National Taiwan University, Taipei 10617, Taiwan
4. School of Atmospheric Sciences, and Guangdong Province Key Laboratory for Climate Change and Natural Disaster Studies, Sun Yat-sen University, Zhuhai 519082, China
5. Southern Marine Science and Engineering Guangdong Laboratory (Zhuhai), Zhuhai 519082, China
6. Woods Hole Oceanographic Institution, Woods Hole, MA 02543, USA
7. Oceanography Division, Naval Research Laboratory, Stennis Space Center, MS 39529, USA
* Correspondence: ipun@ncu.edu.tw

**Abstract:** On 23 August, 2017, Typhoon Hato rapidly intensified by 10 kt within 3 h just prior to landfall in the city of Macau along the South China coast. Hato's surface winds in excess of 50 m s$^{-1}$ devastated the city, causing unprecedented damage and social impact. This study reveals that anomalously warm ocean conditions in the nearshore shallow water (depth < 30 m) likely played a key role in Hato's fast intensification. In particular, cooling of the sea surface temperature (SST) generated by Hato at the critical landfall point was estimated to be only 0.1–0.5 °C. The results from both a simple ocean mixing scheme and full dynamical ocean model indicate that SST cooling was minimized in the shallow coastal waters due to a lack of cool water at depth. Given the nearly invariant SST in the coastal waters, we estimate a large amount of heat flux, i.e., 1.9k W m$^{-2}$, during the landfall period. Experiments indicate that in the absence of shallow bathymetry, and thus, if nominal cool water had been available for vertical mixing, the SST cooling would have been enhanced from 0.1 °C to 1.4 °C, and sea to air heat flux reduced by about a quarter. Numerical simulations with an atmospheric model suggest that the intensity of Hato was very sensitive to air-sea heat flux in the coastal region, indicating the critical importance of coastal ocean hydrography.

**Keywords:** Typhoon; SST cooling; shallow water; vertical mixing; rapid intensification

## 1. Introduction

Typhoon Hato rapidly intensified to a category-3 intense typhoon (increased 10 kt within 3 h) just before making landfall in Macau along the South China coast on 23 August 2017. With 1-min maximum sustained surface winds of 100 kt (≈51.4 m s$^{-1}$) based on the US Joint Typhoon Warning Center (JTWC), Hato severely devastated the city, with 10 people being killed and an economic loss estimated to be over US$ 1.5 billion, according to the Statistics and Census Service of Macau (www.dsec.gov.mo). The damage and impacts were unprecedented in the city's history. Based on the local government record, Hato was the worst typhoon to hit the city for half a century. The deadly outcome was in part due to the limited precautionary measures implemented for the incoming typhoon. Residents were not alerted or aware of the rapidly-intensifying typhoon, especially during the landfall period.

Therefore, it is imperative to understand what caused such an unexpected fast intensification of this landfalling typhoon.

Typhoon intensification processes are extremely complex, involving a wide range of interactions between the atmosphere and the ocean [1–9]. To date, even with many sophisticated operational forecasting systems, to accurately predict the intensification rate of a typhoon, and especially rapid intensification, is still a huge challenge [10–14]. The complete physical mechanisms behind typhoon intensification are still unclear. However, one important and widely-accepted feedback mechanism in typhoon intensity change is typhoon self-induced sea surface temperature (SST) cooling [6,15–20]. A small decrease in SST would significantly affect the energy transfer from the ocean to the typhoon, having important implications for typhoon intensity [7,16,21]. Under the typhoon regime, the cooling of SST is mainly due to wind-driven ocean vertical mixing [22,23]. As a result, the magnitude of SST cooling is closely linked to an initial upper-ocean thermal structure, typically from the surface to 200 m [6,9,19,22,24–26]. Interestingly, it was found that before landfall, Typhoon Hato rapidly intensified from category 2 to category 3 over the shallow water area of the continental shelf along the southern coast of China, where the water depth is generally less than 100 m (Figure 1). Under the strong winds of the typhoon, the coastal shallow waters would be well vertically mixed [27–29]. The primary SST cooling mechanism due to the entrainment of vertical ocean mixing might therefore not be applicable over this shelf area.

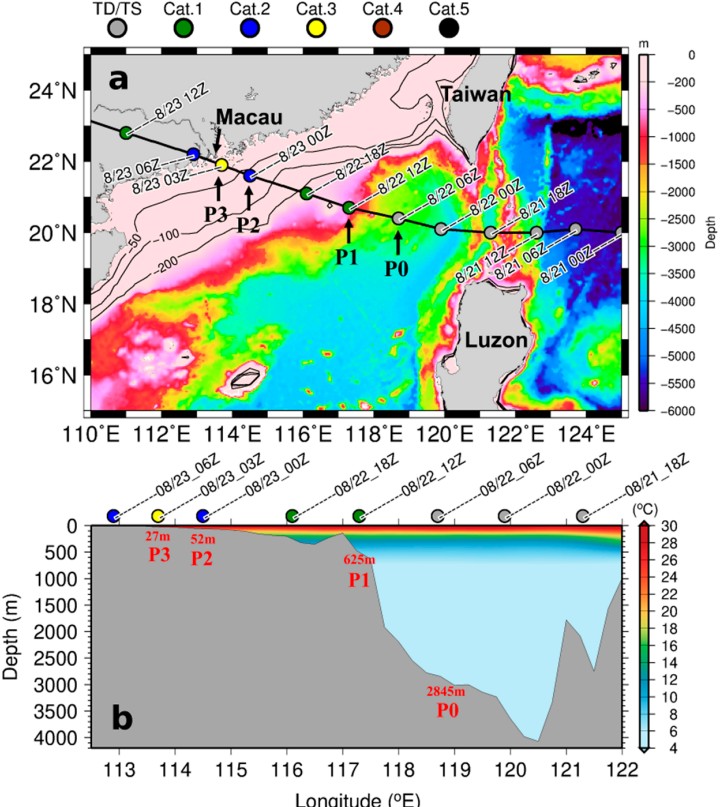

**Figure 1.** (**a**) Bathymetry of the South China Sea based on NOAA/ETOPO2v2 dataset(https://www.ngdc.noaa.gov/mgg/global/etopo2.html). Hato's track is overlaid with color-coded Saffir-Simpson typhoon scale. P0, P1, P2 and P3 depict the selected track points of 06Z 22, 12Z 22, 00Z 23 and 03Z 23 August, respectively. They are used for analyzing SST cooling and heat fluxes at different water depths. The contours indicate the bottom depths of 50, 100 and 200 m. The geographic location of Macau is also indicated. (**b**) The ocean cross section under Hato's track between 18Z 21 and 06Z 23 August 2017 over the SCS. The ocean depths for the four selected points are shown. The background color represents the SCSPOD14 temperature climatology in August.

Emanuel [16] proposed that SST cooling induced by a typhoon could be negated over warm shallow waters due to the lack of deeper cold water, and thus, may affect subsequent intensity changes of typhoons. Based on satellite remote sensing, reductions in SST cooling were observed over the coastal waters along the Florida and east coast of the US after storm passages [30,31]. Using unique in situ observations, Potter et al. [29] recently found that Hurricane Henry, which severely damaged the south coast of the US in 2017, intensified rapidly when it crossed the shallow shelf in the Gulf of Mexico. The observations showed that the warm water extended to the sea bottom and little SST cooling was generated in the shallow water. In fact, the hypothesis of reduced SST cooling in shallow waters has seldom been tested in a real typhoon case undergoing rapid intensification. In this study, we will use oceanic and atmospheric model simulations to investigate whether the rapid intensification of Hato prior to landfall in Macau was linked to the shallow water depth, over which SST cooling by vertical mixing can be greatly suppressed.

## 2. Data and Methodology

### 2.1. Typhoon, Ocean and Atmospheric Datasets

Hato's track data are based on the best warning track from the collaboration site of the JTWC (https://pzal.ndbc.noaa.gov/collab/login.php) that provides the position, intensity and radius of maximum wind (RMW). More importantly, unlike the regular 6-hourly best-track archive, it provides higher temporal (i.e., 3-hourly) information during the critical landfalling period. Based on the JTWC track data, Hato suddenly further intensified during the last 3 h (00Z–03Z 23 August) before making landfall in Macau, although it had been undergoing a rapid intensification after entering the South China Sea (SCS). More strikingly, the intensification rate in the last 3 h prior to landfall doubled, as compared to the previous 24 h (Figure 2). In this study, the intensification rate and translation speed of Hato are computed based on two adjacent track points.

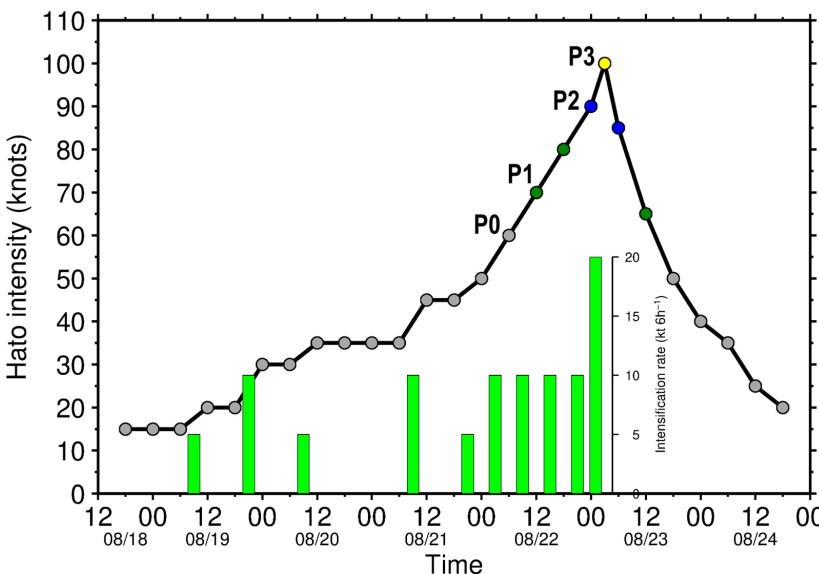

**Figure 2.** Hato's intensity evolution from JTWC track data. The color dots indicate the Saffir-Simpson typhoon scale shown in Figure 1. The green vertical bars indicate the intensification rate in knots per 6 h, computed based on two adjacent track points. Note that only positive rates are shown. The four selected points for numerical experiments are depicted.

High-quality SST observations along the coastal area of Macau and over the SCS are required to examine the ocean environment for the intensification of Hato. In this study, a fine-resolution (4 km) daily SST dataset from the Moderate Resolution Imaging Spectroradiometer (MODIS; https://oceandata.sci.gsfc.nasa.gov/MODIS-Aqua/Mapped/) was used to characterize the ocean surface

conditions prior to Hato's passage. The advantage of using infrared SST data is that it is less affected by land, which is more suitable for the purpose of the present study. Since the infrared SST is often obscured by clouds, an 8-day composite product was used to reduce the missing data rate (Figure 3a).

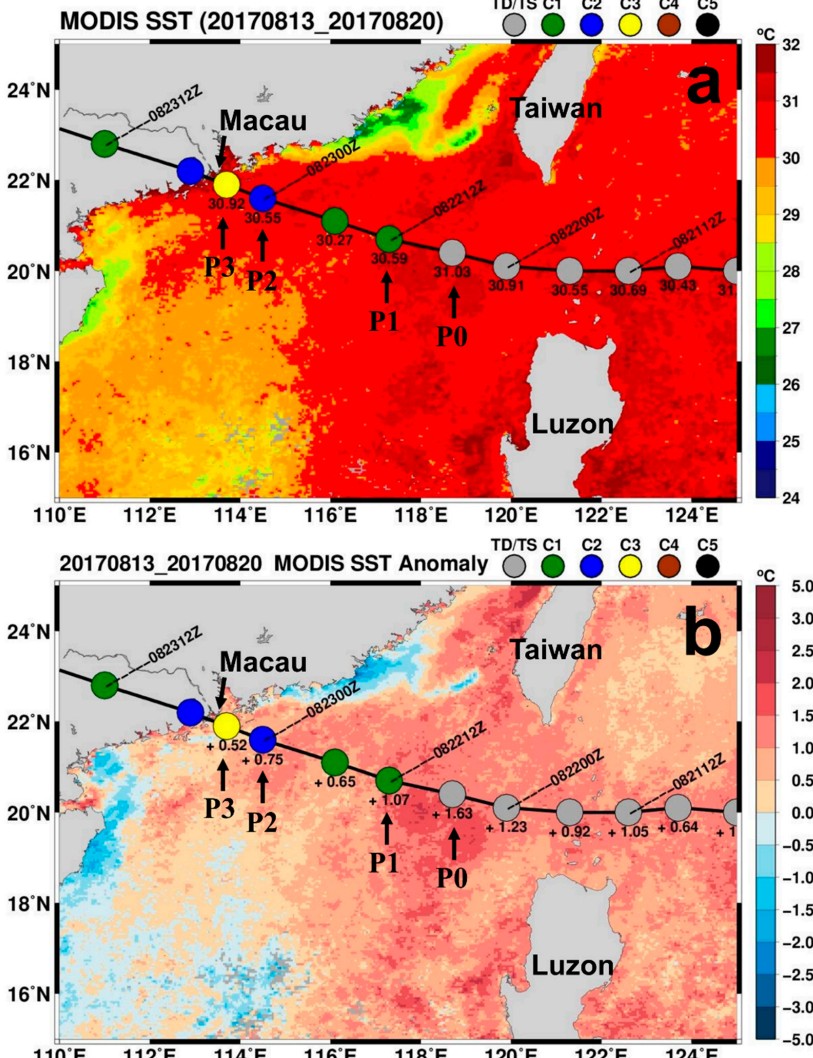

**Figure 3.** (**a**) 8-day SST composite from MODIS from 13 to 20 August. (**b**) The corresponding SST anomaly with respect to the long-term mean SST of August during 2002–2016. The numbers along Hato's track represent the underlying SST and anomaly values in °C. The selected points for numerical experiments are also shown.

Due to the lack of ocean vertical temperature observation around Hato's passage, 0.25° × 0.25° monthly temperature climatology from South China Sea Physical Oceanographic Dataset (SCSPOD14) [32] was used to reconstruct the temperature profiles. Produced by the South China Sea Institute of Oceanology (SCSIO), SCSPOD14 is a new climatology which was specifically built for the SCS. It makes use of all available in situ temperature profiles during the past century (i.e., 1919-2014), including not only data from the comprehensive US World Ocean Database (WOD) and international Argo project, but also exclusive measurements from SCSIO [32]. Therefore, SCSPOD14 is deemed to be one of the best climatological datasets for the SCS.

To reconstruct the ocean subsurface thermal structure, the mixed layer depth (MLD) and underneath vertical temperature gradient (i.e., dT/dz; hereafter called temperature gradient) are extracted from the SCSPOD14 climatology. It should be noted that MLD here refers to the isothermal layer depth with 0.3 °C threshold criterion [33]. The along typhoon track temperature profiles prior to

Hato's passage were generated by a combination of the SCSPOD14 and the corresponding MODIS observed SST (Figure 3a). Such temperature profiles are herein referred to as synthetic profiles, and will be input into an ocean mixing scheme (introduced in the next section) to investigate the effect of shallow water on Hato's rapid intensification.

For air-sea heat flux calculations, atmospheric conditions from the daily global reanalysis of ERA-interim [34] from the European Centre for Medium-Range Weather Forecasts (ECMWF) was employed. The ERA-interim dataset used is on a 6-hourly basis with 0.25° spatial resolution. Near-surface air and dew-point temperatures were extracted for heat flux calculations.

In addition, to represent the most realistic atmospheric condition for Hato's landfalling period, the station observations of air and dew-point temperatures at the Macau Meteorological and Geophysical Bureau were used instead of the ERA-interim at the landfalling point.

## 2.2. Numerical Models

To investigate the effect of suppressed SST cooling in shallow water on Hato's rapid intensification, we will examine SST cooling response and associated heat fluxes at four selected track points, i.e., P0, P1, P2 and P3, as shown in Figure 1. These points are chosen to represent different water depths and typhoon intensities as Hato approached the city of Macau. As listed in Table 1, P3 is the location of Hato just before landfall at 03Z 23 August, with a peak intensity of 100 kt and shallowest water depth of 27 m; P2 is at 00Z 23 August with an intensity of 90 kt and water depth of 52 m; P1 is at 12Z 22 August with an intensity of 70 kt and water depth of 625 m; and P0 is at 06Z 22 August with an intensity of 60 kt and water depth of 2845 m. The water depths along Hato's track over the SCS are depicted in Figure 1b.

**Table 1.** Times, water depths, intensities, radius of maximum wind (RMW) and translation speeds (Uh) of the four selected Hato track points: P0, P1, P2 and P3. Their geographic locations are shown in Figure 1. These track points are chosen for the Price 2009 model experiments.

|  | Time | Water Depth (m) | Intensity (kt) | RMW (km) | Uh (m s$^{-1}$) |
|---|---|---|---|---|---|
| P0 | 06Z 22 August | 2845 | 60 | 18.5 | 6.0 |
| P1 | 12Z 22 August | 625 | 70 | 18.5 | 6.9 |
| P2 | 00Z 23 August | 52 | 90 | 18.5 | 8.1 |
| P3 | 03Z 23 August | 27 | 100 | 18.5 | 8.3 |

In this study, the critical SST values under Hato will be estimated from mixing depth by the ocean mixing scheme proposed by Price [28], which was developed based on the three-dimensional Price-Weller-Pinkel (3DPWP) [35] ocean model:

$$\frac{g\delta\rho d}{\rho_0\left(\frac{\tau}{\rho_0 d}\frac{4R}{U_h}S\right)^2} \geq 0.65 \tag{1}$$

where $g$ is the acceleration due to gravity, $\delta\rho$ the density difference at the base of the mixed layer, $d$ the mixed layer thickness, $\rho_0$ the density of sea water taken as 1024 kg m$^{-3}$, $R$ and $U_h$ are the maximum wind radius and translation speed of the typhoon, and $S$ is the scale parameter taken as 1.3, as proposed by Price [28] to account for the earth rotation effect; finally $\tau$ is the wind stress induced by the typhoon, computed via:

$$\tau = \rho_a C_D V^2 \tag{2}$$

where $\rho_a$ is the density of air taken as 1.2 kg m$^{-3}$, $C_D$ the drag coefficient, which is based on Powell et al. [36] accounting for high-wind condition, and $V$ is the maximum surface wind, which directly adopts the surface wind speed from JTWC. Hereafter, this scheme is referred to as the Price 2009 model.

Basically, this model is intended to find the thickness (or depth, *d*) from the sea surface until the left-hand side of the Equation (1) is equal or greater than the bulk Richardson number of 0.65 [37]. The Price 2009 model is an efficient way to estimate typhoon mixing depth, i.e., *d*, by a given typhoon's general characteristics (i.e., *V*, *R* and $U_h$) and ocean stratification [19,38]. The SST cooling generated by the typhoon was obtained by vertically averaging the temperatures within the mixing depth *d*. In this study, the corresponding ocean temperature profiles input to the Price 2009 model were reconstructed by the combination of MODIS observations and SCSPOD14 climatological ocean thermal structure, as described in Section 2.1. Figure 4 shows the monthly variability of the MLD and temperature gradient at four selected locations based on the SCSPOD14. It can be seen that the MLD and temperature gradient vary considerably between summer (May to September) and winter (October to March), in particular at the shallow water area (P2 and P3); this is probably due to enhanced surface heating in summer and strong wind mixing in winter. In this study, to account for the possible variability of the MLD and temperature gradient, the annual mean values were used to reconstruct the temperature profiles for the Price 2009 model experiments. Figure 5 shows the synthetic profiles for the four selected points prior to Hato's passage. It is noteworthy that at shallow water locations P2 and P3, the synthetic temperature profiles extend until they reach the bottom depth (blue and red curves in Figure 5). For all the Price 2009 model experiments, salinity remains at a constant of 33 psu throughout the water column.

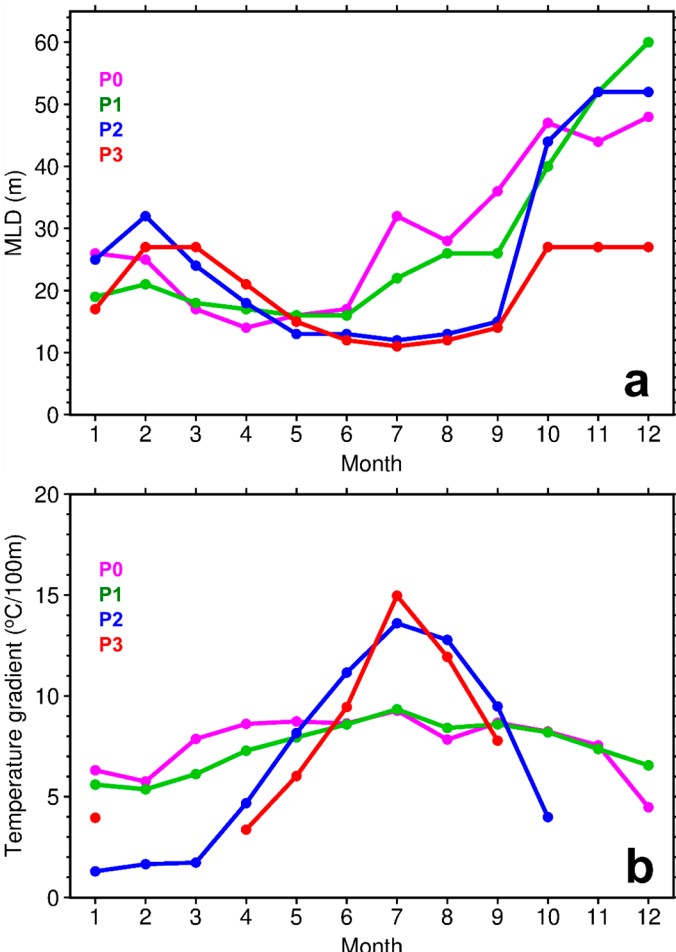

**Figure 4.** Monthly (**a**) MLD and (**b**) vertical temperature gradient at the four selected locations based on the SCSPOD14 climatology. Note that in some winter months there is no temperature gradient at P2 and P3 due to the whole water column is completely mixed. The locations of P0, P1, P2 and P3 are depicted in Figure 1.

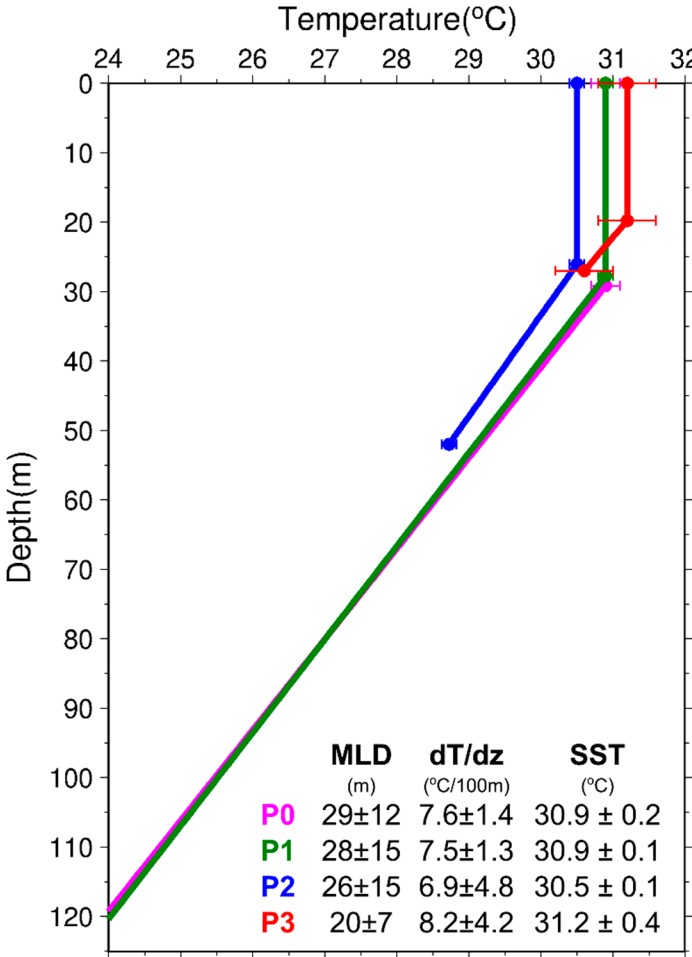

**Figure 5.** Synthetic temperature profiles for P0 (magenta), P1 (green), P2 (blue) and P3 (red), which are reconstructed by MODIS observed SST (Figure 3a) and annual mean MLD and *dT/dz* from SCSPOD14. The mean values and standard deviations of MLD, *dT/dz* and SST are shown. The horizontal error bars represent the standard deviation of SST within RMW. The locations of P0, P1, P2 and P3 are depicted in Figure 1. These profiles are input to the Price 2009 model to assess the shallow water effect on SST cooling and heat flux supply to Hato.

In addition to the Price 2009 model, a sophisticated full ocean model, the Luzon Strait Nowcast/Forecast System (LZSNFS) [39–45], developed by the US Naval Research Laboratory (NRL), was used to examine the ocean response to typhoon Hato. The LZSNFS model deals with all three-dimensional ocean processes, including coastal and bottom boundary layer effects that the Price 2009 model does not account for. Therefore, it would provide more thorough insights into the SST cooling in the shallow water area. Furthermore, to explore the impact of shallow water on Hato's rapid intensification, the advanced hurricane version of the Weather Research and Forecasting (WRF) model, hereafter AHW, was employed to perform a series of numerical experiments on Hato's intensity. The descriptions of the LZSNFS and AHW models will be introduced later in the corresponding sections.

*2.3. Air-Sea Enthalpy Flux*

Based on the under-typhoon SST estimated from the Price 2009 model, the corresponding sensible ($Q_S$) and latent ($Q_L$) heat fluxes, or total heat fluxes, can be calculated through the bulk aerodynamic formulas [21,46,47]:

$$Q_S = C_H V (T_s - T_a) \rho_a C_{pa} \tag{3}$$

$$Q_L = C_E V (q_s - q_a) \rho_a L_{va} \tag{4}$$

where $C_H$ and $C_E$ are the exchange coefficients for sensible and latent heat, both taken as $1.3 \times 10^{-3}$ based on measurements under high wind condition [48], $T_s$ and $T_a$ the SST and near-surface air temperature, $q_s$ and $q_a$ the surface and air specific humidities calculated based on $T_s$, $T_a$ and dew-point temperature, and $C_{pa}$ and $L_{va}$ the air heat capacity and latent heat of vaporization, respectively. Throughout this paper, heat fluxes that are transferring from the ocean to the atmosphere are defined as positive.

## 3. Warm SST and Shallow Water Depth

### 3.1. Observations

As the ocean is an energy source to fuel the intensification of a typhoon [1,4,16,17,19,26,46,49,50], it is natural to speculate about the role the ocean played in Hato's rapid intensification. The high-resolution MODIS SST composite between 13 and 20 August shows that the SCS was abnormally warm in 2017 with the pre-typhoon SST nearly 31 °C ahead of Hato's passage (Figure 3a), which is about 1 °C warmer than climatology (2002–2016) (Figure 3b). Indeed, 2017 had the highest mean SST (30.2 °C) over the northern South China Sea (15–23 °N, 110–120 °E) since 2010 for the same period during 13–20 August (not shown). Cheng and Zhu [51] also reported that 2017 was the warmest year on record in terms of global ocean heat storage at between 0–2000 m. In addition, Hato was the first strong typhoon that appeared in the SCS in 2017, so the summertime SST was nearly undisturbed and remained at higher values (~31 °C). Such a warm ocean environment probably facilitated the development of Hato.

It is well-known that a typhoon acts to cool the SST below, and that such SST cooling is considered an important negative feedback mechanism in controlling typhoon intensity [6,15,16,18,19,22,26,52,53]. In other words, the abnormally warm SST alone may not be enough to explain the enhanced intensification before landfall (Figure 2). The magnitude of the SST cooling induced by Hato therefore becomes critical.

It is important to note that while Hato's rapid intensification took place along the slope of the continental shelf since 06Z 22 August, the rate suddenly accelerated in the coastal area near Macau, where the water depth is only 30–50 m (Figures 1 and 2). The rapid change in intensification rate appears to be coincident with the shoal of the bathymetry. Recently, several studies have proposed that in the relatively warm shallow water regions, the typhoon-induced SST cooling due to vertical mixing may be prohibited, because of limited supply of deeper cold water [3,16,28,29,31]. Therefore, it is of great interest to investigate whether the rapid intensification of landfalling Typhoon Hato was related to the shallow water depth.

Figure 5 compares the synthetic temperature profiles from far deep ocean (~3000 m) to the coast of Macau (~30 m) before the passage of Hato. Again, the profiles were obtained by MODIS observed SST with climatological MLD and temperature gradient from SCSPOD14. At the maximum intensity point (100 kt, category-3) at 03Z 23 August before landfall, the water depth was only about 27 m (red profile in Figure 5). Under this circumstance, no matter how strong the typhoon winds, the maximum depth at this point to which winds could mix would be 27 m. Furthermore, although the mixed layer depth was comparatively shallower as compared to the deep ocean points (i.e., P0 and P1), the high SST (31.2 °C) and shallow water depth (27 m) resulted in very warm bottom water (30.6 °C) at P3. In other words, the overall water temperature at P3 was greater than 30.6 °C. This would obviously have great implications for SST cooling.

### 3.2. Price 2009 Model Simulations

To demonstrate the shallow water effect on SST cooling on Hato's intensification, four specific track points of Hato with different water depths and wind speeds were chosen to carry out a series of numerical experiments using the Price 2009 model. The four selected track points from deep to shallow waters are 06Z 22, 12Z 22, 00Z 23 and 03Z 23 August, referred to as P0, P1, P2 and P3, respectively. Their locations, water depths and corresponding typhoon characteristics are shown in Figure 1 and Table 1. P0 and P1 are the deep ocean group located in the open SCS with water depths of 2845 m and

625 m (Figure 1b). The wind speeds at these two points were 60 kt and 70 kt, respectively. P2 and P3 are the shallow water group. P2 is located on the shelf slope with water depth of 52 m and wind speed of 90 kt, while P3 is the very landfalling point with water depth of 27 m and peak of 100 kt. Based on JTWC data, the radius of maximum wind (RMW) was 18.5 km for all these points. Hato's moving speed at P0 and P1 was around 6–7 m s$^{-1}$; it then sped up to ~8 m s$^{-1}$ at P2 and P3 prior to landfall (Table 1).

The simulated mixing depths and SST cooling values from the Price 2009 model are shown in Figure 6 and Table 2. Given the initial temperature profiles and Hato's characteristics (i.e., *V*, *R* and $U_h$ in Table 1), the mixing depths output by the Price 2009 model were 56, 63, 52, and 27 m at P0, P1, P2 and P3, respectively. It was shown that the entire water column over the shallow water area (P2 and P3) was completely mixed from top to bottom as forced by Hato (Figure 6a). Because of the shallow depth and lack of deeper cold water, little SST cooling (0.1–0.5 °C) was generated over the shallow water by vertical mixing (Figure 6a and Table 2). Furthermore, due to the pre-existing anomalously warm SST, the under-Hato SST was preserved at a high value, i.e., up to 31.1 °C, near the coast of Macau (Figure 6a). In contrast, P0 and P1 in the deeper open SCS had relatively deep typhoon mixing depths (Figure 6b). However, the SST cooling values at these two points were moderate, i.e., 0.5 °C and 0.7 °C for P0 and P1 (Table 2), respectively. This is due to the fact that the warmer subsurface compensated for the deeper typhoon mixing depth at these two locations (Figure 6b).

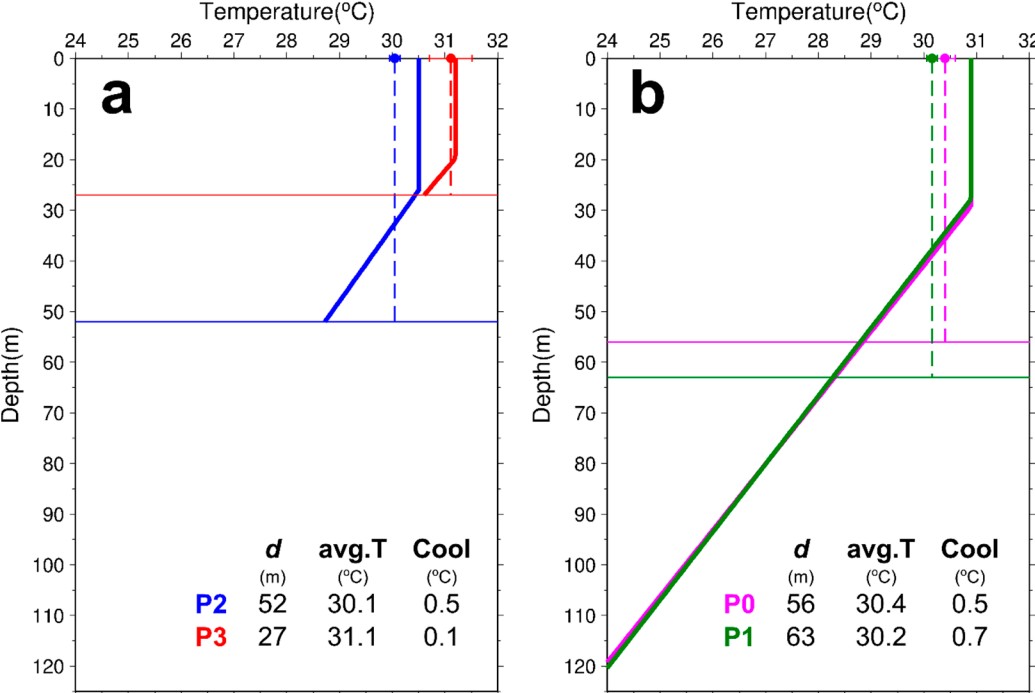

**Figure 6.** Vertical temperature mixing results from the Price 2009 model for (**a**) shallow water points (i.e., P2 and P3) and (**b**) deeper water points (i.e., P0 and P1). The horizontal lines represent the mixing depths, *d*. The vertical dashed lines represent the average temperatures within the mixing depths. Note that the mixing is all the way down to the sea bottom on the continental shelf area (**a**). The values for estimated typhoon mixing depth, under-typhoon SST (i.e., vertically average temperature) and cooling are shown in the lower-left of each plot. The error bars indicate the SST standard deviation within RMW. The locations of P0, P1, P2 and P3 are depicted in Figure 1.

**Table 2.** Estimates of mixing depth, vertically averaged temperature (i.e., under-typhoon SST), and SST cooling at four selected points by the Price 2009 model.

|  | Time | Mixing Depth (m) | Averaged Temperature (°C) | SST Cooling (°C) |
|---|---|---|---|---|
| P0 | 06Z 22 August | 56 | 30.4 | 0.5 |
| P1 | 12Z 22 August | 63 | 30.2 | 0.7 |
| P2 | 00Z 23 August | 52 | 30.1 | 0.5 |
| P3 | 03Z 23 August | 27 | 31.1 | 0.1 |

To evaluate the impact of ocean mixing on the energy supply to Hato, the air-sea enthalpy flux at each selected track point was calculated and compared (Table 3). At the deeper open SCS of P0 and P1, the total (sensible + latent) heat fluxes were moderate, i.e., 762 and 785 W m$^{-2}$, respectively. Most of the enthalpy flux was due to latent heat. Due to small SST cooling over the shallow water and increasing wind speed, the total heat flux increased to 1203 W m$^{-2}$ at P2 when Hato entered the shelf water at 00Z 23 August. More importantly, because of the nearly invariant high SST (31.1 °C) facilitated by the shallow water constraint, there was a large amount of total heat flux of 1927 W m$^{-2}$ at 03Z 23 August (P3). This suggests that energy from the ocean was continuously feeding into Hato as it was making landfall. Due to the shallow water effect, as well as the higher wind speed, the heat flux at P3 was 153%, 145% and 60% more than P0, P1 and P2, respectively. Most strikingly, this amount of heat flux was about 68% more than the heat flux available for Supertyphoon Haiyan (2013) at the similar strength (~1150 W m$^{-2}$) estimated by Lin et al. [46], who applied the same bulk aerodynamic formulas (Section 2.3). It is noteworthy that the change in the intensification rate is roughly in line with the heat flux variation at these four selected points (Table 3 and Figure 2). This result suggests that such a large amount of heat flux would contribute to the rapid intensification of landfalling Hato.

**Table 3.** Air-sea enthalpy fluxes [sensible ($Q_S$) + latent ($Q_L$)] (unit: W m$^{-2}$) at four selected track points of Hato. The fluxes are computed based on the bulk aerodynamic formulas (i.e., Equations (3) and (4)) with consideration of SST cooling (Table 2) due to vertical mixing.

|  | Time | $Q_S$ | $Q_L$ | $Q_S + Q_L$ |
|---|---|---|---|---|
| P0 | 06Z 22 August | 143 | 619 | 762 |
| P1 | 12Z 22 August | 139 | 646 | 785 |
| P2 | 00Z 23 August | 242 | 961 | 1203 |
| P3 | 03Z 23 August | 370 | 1557 | 1927 |

*3.3. Experiments for No Depth Restriction and SST Warming*

To further assess the impact of shallow water effect on SST cooling, the depth restriction is relaxed for P3 and P2 points in the Price 2009 model (Figure 7a). As shown in Table 4, the mixing depths for P3 and P2 would deepen to 68 and 69 m, respectively, if there were no bathymetry. This is equivalent to 152% and 33% increases in typhoon mixing depth as compared to the results with real ocean depths. More importantly, because of the deeper mixing depth and the availability of deeper cold water, the SST cooling would be 1300% (from 0.1 to 1.4 °C) and 80% (from 0.5 to 0.9 °C) stronger at P3 and P2 (Figure 7a and Table 4). Therefore, SST cooling at the landfalling point (P3) would become the largest among all the testing points (Table 4). The increases in SST cooling led to significant drops in air-sea heat fluxes, especially at P3. Without the shallow water depth, the heat flux would have reduced by 25% at P3, and 12% at P2 (Table 4).

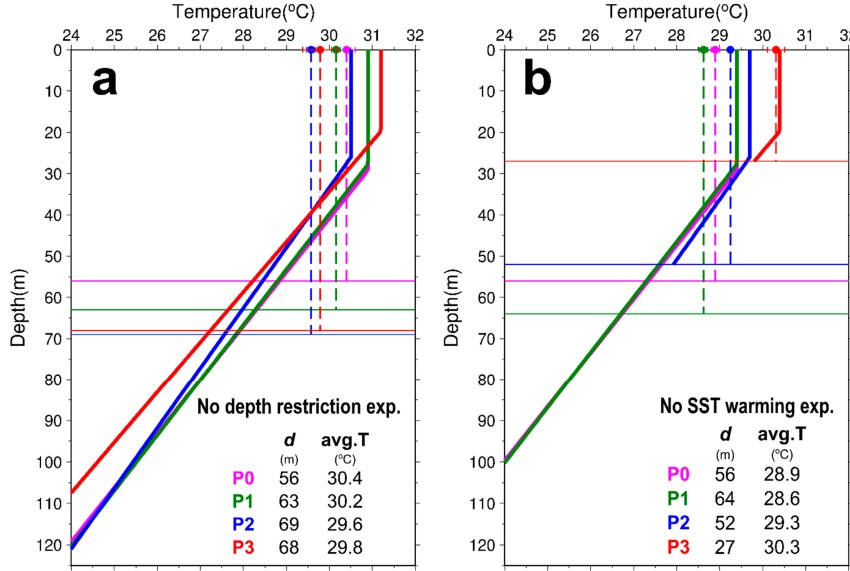

**Figure 7.** (**a**) No depth restriction experiment and (**b**) No SST warming experiment with the Price 2009 model. The horizontal lines represent the mixing depths, *d*. The vertical dashed lines represent the average temperatures within the mixing depths. The error bars indicate one standard deviation of SST within RMW. The values for mixing depths and average temperatures are shown in each plot, while the locations of P0, P1, P2 and P3 are depicted in Figure 1.

**Table 4.** Mixing depths, SST cooling values and the corresponding enthalpy fluxes at four selected points in no depth restriction experiment. Note that the SST cooling generally increases with the mixing depth.

| | Time | Mixing Depth (m) | Cooling (°C) | $Q_S+Q_L$ (W m$^{-2}$) |
|---|---|---|---|---|
| P0 | 06Z 22 August | 56 | 0.5 | 762 |
| P1 | 12Z 22 August | 63 | 0.7 | 785 |
| P2 | 00Z 23 August | 69 | 0.9 | 1054 |
| P3 | 03Z 23 August | 68 | 1.4 | 1451 |

As the SST was unusually warm in 2017, it is also interesting to assess the impact of warming SST on the heat fluxes. To do that, the long-term mean SST in August based on MODIS observations during 2002–2016 was used to construct the synthetic temperature profiles (Figure 7b). It should be noted that there was nearly no change in mixing depth and SST cooling in this experiment, due to the same ocean thermal structure (i.e., SCSPOD14). The results show that without the SST warming, the heat fluxes would have reduced by 42%, 47%, 21% and 15% for P0, P1, P2 and P3 (Table 5), respectively. According to these two numerical experiments, it is suggested that the warm SST, and especially the reduced cooling over shallow water (due to less cool, deep water), likely played crucial roles in Hato's rapid intensification just prior to its landfall.

**Table 5.** Mixing depth, averaged temperature and enthalpy fluxes at four selected track points in no SST warming scenario. Namely, long-term mean SST between 2002 and 2016 is used instead of MODIS observed SST in synthetic temperature profiles (Figure 7b).

| | Time | Mixing Depth (m) | Averaged Temperature (°C) | $Q_S + Q_L$ (W m$^{-2}$) |
|---|---|---|---|---|
| P0 | 06Z 22 August | 56 | 28.9 | 445 |
| P1 | 12Z 22 August | 64 | 28.6 | 413 |
| P2 | 00Z 23 August | 52 | 29.3 | 951 |
| P3 | 03Z 23 August | 27 | 30.3 | 1635 |

*3.4. LZSNFS Simulation*

Although the Price 2009 model offers an efficient way to estimate SST cooling induced by a typhoon (e.g., [19,38,54]), it might be too idealized to deal with the complex nature of typhoon-ocean interaction in a coastal environment (e.g., [27,47]). Given this concern, a simulation for Hato was performed, applying a full three-dimensional dynamic ocean model with realistic topography and forced with real-time meteorological forcing. The model used was the Luzon Strait Nowcast/Forecast System (LZSNFS), a high-resolution coastal model based on the real-time Ocean Nowcast/Forecast System developed by NRL [55,56]. LZSNFS was mainly used to study the nonlinear internal tides/waves in northern SCS [40–45,57], but also to study the impacts of typhoons [39]. The model has 40 hybrid vertical layers, and for this study, the model domain was extended farther to the west to cover all of Hato's track in SCS with an improved horizontal resolution of 1.3 km. The surface forcing comes from higher resolution (0.15°) Coupled Ocean/Atmosphere Mesoscale Prediction System (COAMPS). The open boundary conditions are from global Hybrid Coordinate Ocean Model (HYCOM). The initial ocean condition is based on the Modular Ocean Data Assimilation System (MODAS), which predicts three-dimensional ocean temperatures with satellite sea surface height and SST observations. It should be noted that the Price 2009 model uses the synthetic profiles derived from the MODIS SST and climatological ocean structure. This aims to provide independent results, since models contain uncertainty. LZSNFS also includes tidal currents, which are driven by the tidal forcing from the Oregon State University (OSU) [58]. In addition, river runoffs are also taken into account in this model. The input discharge data are received from the Global Runoff Data Centre (GRDC) of the World Meteorological Organization (WMO). For the simulation of Hato, the model was started from 1 May 2017 to the end of year; outputs from 14–25 August 2017 were analyzed.

Figure 8a–c shows LZSNFS-simulated SSTs before, during and after Hato travelled over the continental shelf. It is important to note that there was nearly no SST decrease on the shelf, especially where the water depth was less than 50 m. The SST of the shelf area remained relatively warm, even two days after the Hato's passage (Figure 8c). The reduced SST cooling can be clearly seen on the difference map between 12Z 21 August and 03Z 23 August, the time when Hato was about to make landfall (Figure 8d).

To obtain a better insight of temperature evaluation, hourly vertical temperature structures at these four selected points were extracted from the LZSNFS simulation (Figure 9). It is interesting and important to note that in the NRL model, the coastal water (i.e., P3) was warm and uniform, even before the typhoon, while the temperature reduced little during and after. This indicates that the mixing due to strong winds did not change the SST on the shallow shelf near Macau because of lack of cooler deeper water (Figure 9a). Similarly, the upper 20 m temperature changed only a small amount before and after Hato at P2, and the water column became fully mixed afterward (Figure 9b). On the other hand, it was found that semi-diurnal internal tides/waves occurred in the deeper basin (Figure 9c,d). The impact of Hato was mostly in the surface mixing, but it also shows upwelling with isotherms rising almost 10 m.



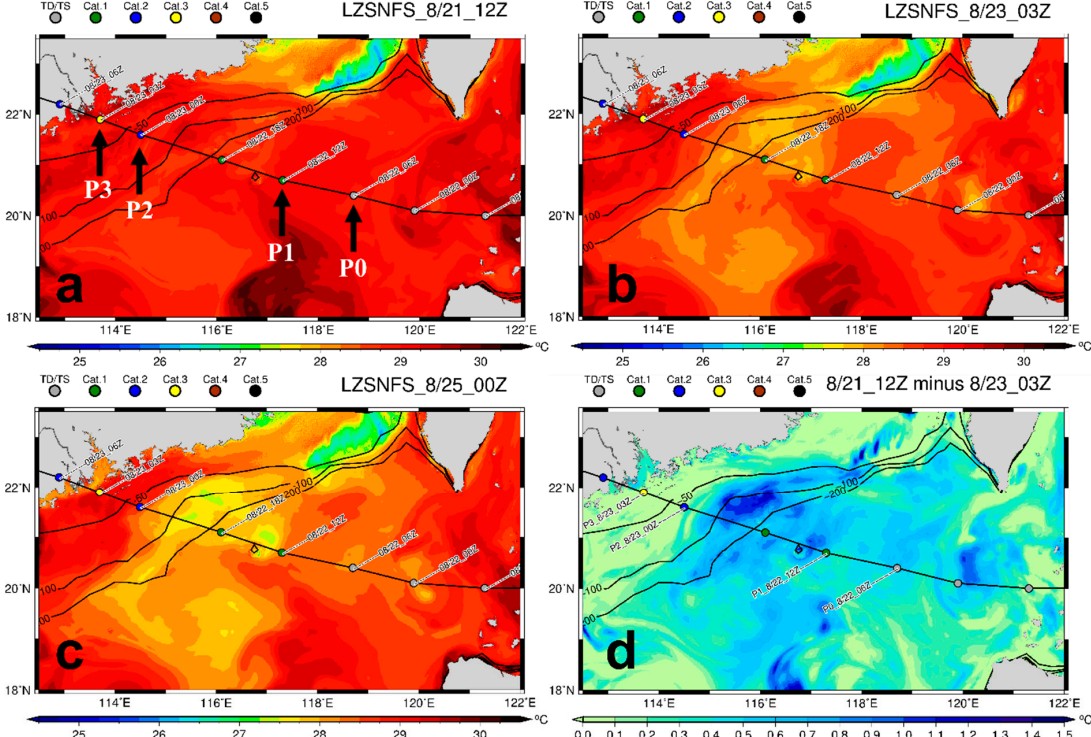

**Figure 8.** LZSNFS simulated SST at (**a**) 12Z 21, (**b**) 03Z 23 and (**c**) 00Z 25 August, corresponding to the situations before, during, and after Hato's passage. (**d**) SST difference map between (**a**) and (**b**), showing the SST cooling during Hato before making landfall on Macau. Hato's track is superimposed with the black contours indicating the bottom depths of 50, 100 and 200 m. The geographic locations of the selected four track points are also indicated in (**a**).

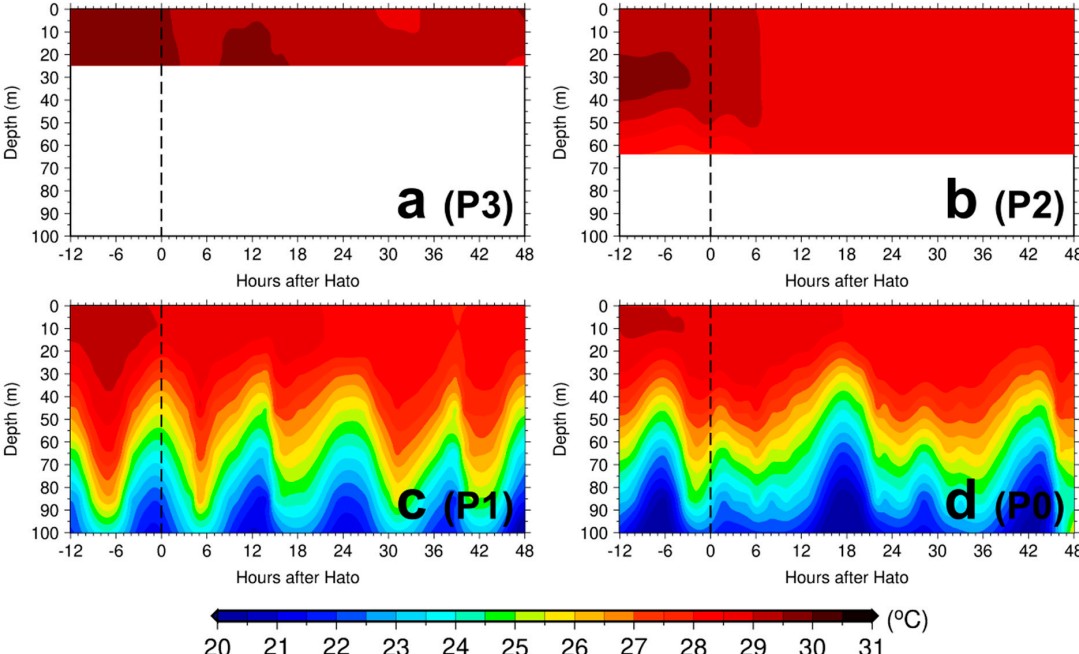

**Figure 9.** Vertical temperature evolutions at (**a**) P3, (**b**) P2, (**c**) P1 and (**d**) P0 from LZSNFS simulations. Vertical dashed line represents the arrival of Hato. Note that all four plots are in the same scale, the ocean depths in (**a**) and (**b**) are less than 100 m.

Figure 10 shows the corresponding SST evolutions between +/−1.5 days of Hato's passage. It was found that the cooling was relatively small in the shallow water area, especially at P3. The SST cooling was assessed for each selected location based on different periods relative to the arrival of Hato (Table 6). In general, the simulated cooling at P3 was merely 0.3 °C (0.5 °C) during (after) Hato, which was consistent with the estimate (0.1 °C) from the Price 2009 model. It should be noted that the cooling in LZSNFS simulations accounts for air-sea heat loss that the Price 2009 model does not consider. Therefore, the stronger cooling in the LZSNFS model is reasonable. The cooling values at P2 and P3 are comparable in LZSNFS simulations (Table 6). This is likely due to the warmer initial ocean structure in the LZSNFS. The result also indicates the importance of initial ocean structure. Away from the coast of Macau, i.e., P0 and P1, the SST cooling values were relatively small, ~0.3 °C and ~0.7 °C for during and after storm, respectively. These values also generally agree with those of the assessment from the Price 2009 model. Nevertheless, the simulation results from LZSNFS provided robust, independent evidence of the cooling suppression effect due to shallow water depth. It also indicated the validity of using the Price 2009 model in this relatively wide continental shelf area, where water temperature tends to be uniform and vertical mixing is the dominant cooling mechanism.

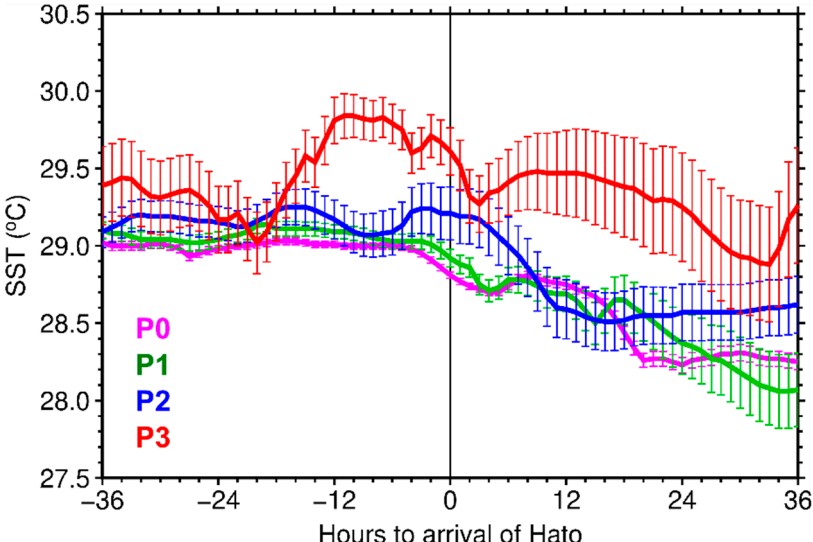

**Figure 10.** SST evolution simulated by LZSNFS at four selected track points. The error bars depict the variation of SST within the RMW (18.5 km), while the black vertical line depicts the arrival of Hato. The corresponding SST cooling values are summarized in Table 6.

**Table 6.** LZSNFS simulated SST (cooling) for before, during and after Hato at the four selected track points. Before, during and after are defined as the mean values from −18 to −6 h, 0 to +12 h, and +18 to +30 h, respectively. Cooling is positive and with respect to the before value. The unit is in °C.

|  | **Before** | **During** | **After** |
|---|---|---|---|
| P0 | 29.0 | 28.8(0.2) | 28.3(0.7) |
| P1 | 29.1 | 28.8(0.3) | 28.4(0.7) |
| P2 | 29.2 | 29.0(0.2) | 28.6(0.6) |
| P3 | 29.7 | 29.4(0.3) | 29.2(0.5) |

## 4. Hato Intensity Simulations

In order to investigate the impact of SST cooling on Hato's rapid intensification, simulations of Hato's intensity with different SST scenarios were performed using the AHW model [59,60]. AHW is the advanced WRF for tropical cyclone simulation purposes with a vortex-following technique. In this study, AHW comprises one big static and two smaller vortex-following domains (Figure 11). The horizontal resolutions are 36, 12 and 4 km from the outermost to the innermost domains with the corresponding

grid points of 121 × 100, 232 × 199 and 250 × 250, respectively. There are 37 vertical levels with the top level at 50 hPa. The time steps are 135s, 45s and 15s, correspondingly. The surface layer physics is based on MM5 similarity [61], microphysics is based on WRF Single-Moment 6-class (WSM 6) [62], planetary boundary layer parameterization is based on Yonsei University (YSU) scheme [63], radiative forcing is based on Iacono et al. [64], and cumulus parameterization is based on Tiedtke scheme [65,66]. Note that the cumulus parameterization was only applied for the outer two domains. The boundary conditions for the atmosphere and SST come from the 6-hourly Climate Forecast System version 2 (CFSv2) operational analysis from the US National Oceanic and Atmospheric Administration (NOAA).

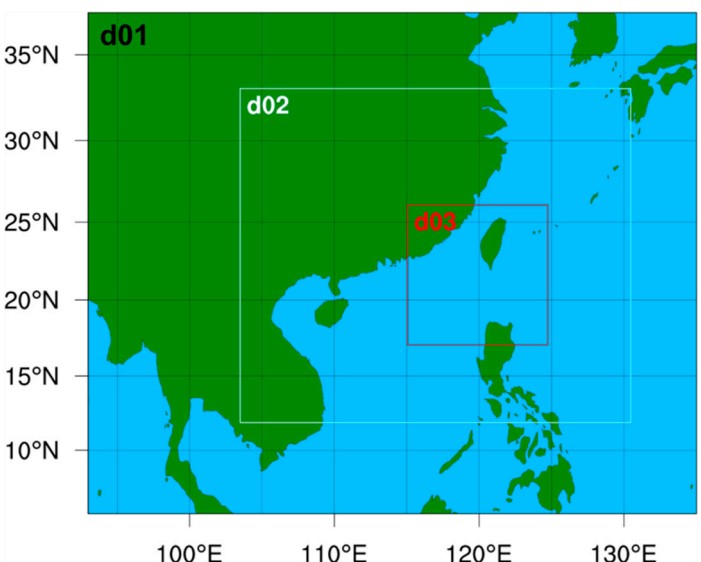

**Figure 11.** Three nesting domains of the AHW model. The horizontal resolutions are 36, 12 and 4 km from the outermost (d01) to the innermost domains (d03) with the corresponding grid points of 121 × 100, 232 × 199 and 250 × 250, respectively. Note that d01 is static and d02 and d03 are vortex-following domains. Domains d02 and d03 shown here are at the time of AHW model initialization.

Since the main focus of this study is the constrained SST cooling effect by the coastal shallow water, Figure 12 shows the SST and cooling maps from CFSv2 before and during Hato's landfall. It was found that the SST of CFSv2 was in good agreement with MODIS observation prior to the passage of Hato (Figure 12a vs. Figure 3a). That is, the SST was unusually warm ahead of Hato over the SCS. However, CFSv2 seems to have a strong SST cooling over the continental shelf along the path of Hato (Figure 12b). That is in contrast to the simulation of LZSNFS model (Figure 8d). Therefore, it is interesting to see how the intensity of Hato responded to the underlying SST in the AHW model.

In this study, two simulation runs of the AHW model were designed to clarify the role of SST (or cooling) in Hato's intensity, especially during the landfall in Macau. The first simulation was with CFSv2 SST updated at every 6 h in AHW model. This simulation is referred to as vSST experiments, as the SST varies over time. The second simulation was with a constant SST, fixed at 12Z 21 August, i.e., the beginning of the simulations; this simulation is referred to as cSST experiments, in which SST does not change over time. In other words, the cSST experiment did not consider any SST cooling induced by the typhoon.

In two AHW experiments, Hato's track was successfully simulated (Figure 13a). Figure 13b compares Hato's intensity between vSST and cSST runs. It was found that Hato's intensity was significantly underestimated in the vSST simulation. The maximum intensity was 11 kt weaker than the JTWC observation (89 kt vs. 100 kt) (blue curve in Figure 13b). In contrast, using the prescribed SST, Hato's intensity and its evaluation were well simulated in cSST; the maximum intensity was 98 kt (red curve in Figure 13b). Comparing these two simulation results, it is evident that Hato's intensity was highly sensitive to the underlying SST, and the underestimation in the vSST experiment was

obviously due to the strong SST cooling in CFSv2 over the shallow water off Macau (Figure 12b). These model results also suggest that the warm SST was maintained over the coastal area due to a lack of deeper cold water, as demonstrated by the Price 2009 model in Section 3.2.

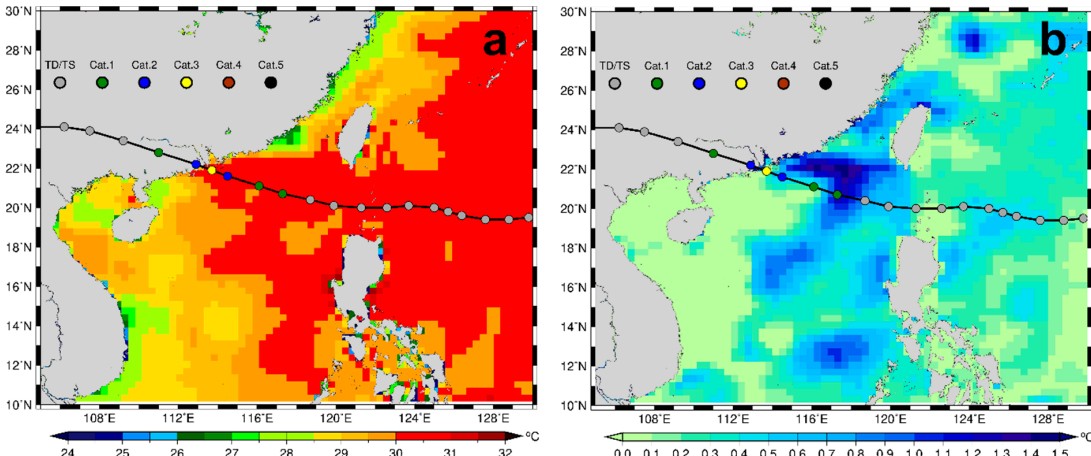

**Figure 12.** (**a**) CFSv2 SST at 12Z 21 August 2017. (**b**) SST cooling in CFSv2 at 03Z 23 August 2017 with respect to (**a**). Hato's track is superimposed.

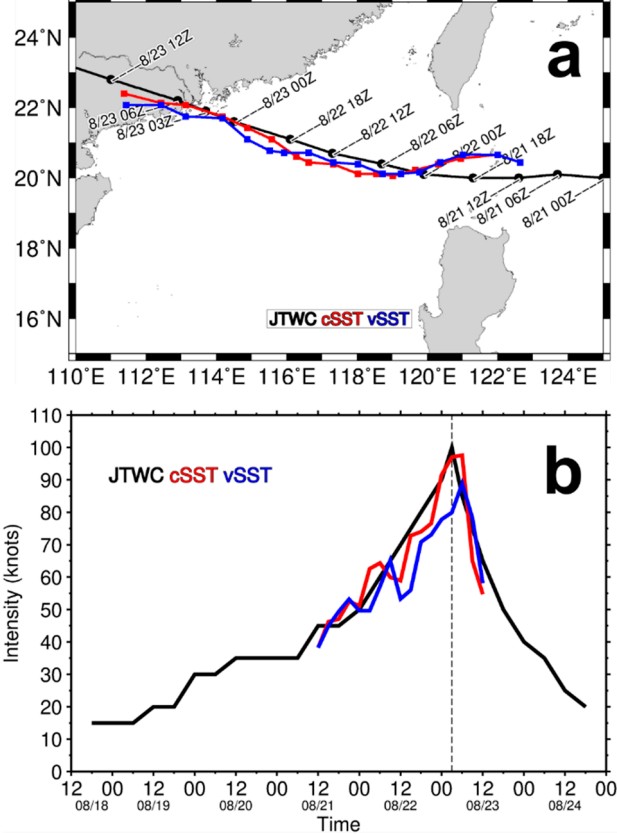

**Figure 13.** (**a**) Comparison of AHW simulated tracks with JTWC observation. (**b**) Hato intensity simulations by AHW. Red curve depicts constant SST experiment (cSST), blue curve depicts variant SST experiment (vSST), and black curve depicts JTWC observed track. The vertical dashed line indicates the landfall of Hato.

Therefore, it can be concluded that the shallow water depth and ocean warming were critical for Hato's rapid intensification.

## 5. Discussion and Conclusion

On 23 August, 2017, Typhoon Hato rapidly intensified by 10 kt within 3 h before making landfall in Macau, where it caused unprecedented damage. This study demonstrates that the coastal bathymetry, as well as the abnormal warming in the South China Sea (SCS), likely played critical roles in the rapid intensification of Hato. The high-resolution MODIS observations revealed that the SST ahead of the passage of Hato, especially along the coast near Macau, was about 1 °C warmer than climatology. In fact, the overall SST over the northern SCS in 2017 was the warmest since 2010. Most importantly, it was found that the shallow water depth (~30 m) off Macau may played a unique role by preventing or minimizing the ocean cooling effect due to typhoon-forced vertical mixing. Based on the Price 2009 model, little SST cooling (0.1~0.5 °C) could be generated by Hato over the shallow water, due to a lack of deeper cold water for the mixing. Such an important shallow water mechanism for Hato-induced SST cooling is further supported by the full ocean model simulation of the LZSNFS.

With the suppressed SST cooling effect, a large amount of heat flux (1927 W m$^{-2}$) from the ocean was readily available for Hato, even though it was making landfall. This study also determined that if no such shallow water existed, the heat flux during the landfall would have reduced by 25% due to enhanced SST cooling. In this sense, such conditions may have hampered Hato's intensification. As evidenced by AHW simulations, the intensity of landfalling Hato was very sensitive to the nearly invariant SST over the continental shelf. The results suggest that the constrained effect of SST cooling would indeed positively feed back to Hato's intensity, allowing Hato to rapidly intensify and eventually become a severe typhoon just 3 h prior to landfall.

Warm, shallow coastal waters are the usual condition during the summer season of southern China. As a result, there may be a possibility for a landfalling typhoon to intensify rapidly or to maintain its intensity. In this situation, it would add more challenges to the already difficult intensity prediction situation, and would pose a greater threat to the people living in that coastal area. However, it should be noted that not all the shallow water areas are capable of increasing typhoon intensification. That also depends on their vertical thermal structure and stratification. Recent studies have indicated that SST could drop dramatically over coastal regions prior to the passage of storms due to strong stratification [47,67]. Therefore, there is a pressing need to set up an operational temperature observing array in the continental shelf along the northern South China Sea, over which several typhoons pass by each year. The continuous monitoring of coastal water temperatures would improve our understanding and prediction abilities of rapid intensification of incoming typhoons. Nevertheless, Typhoon Hato provides such a vivid example of the potential of this kind of typhoon, and more in-depth studies using a fully coupled model are needed in the future.

**Author Contributions:** I.-F.P., J.C.L.C. and I.-I.L. initiated and designed the study. I.-F.P. conducted the numerical experiments and wrote the original draft. K.T.F.C. conducted AHW simulations. J.F.P. consulted on the use of the Price 2009 model and discussed the results. D.S.K. conducted LZSNFS simulations. C.-C.L., Y.-L.W. and H.-C.H. contributed to data analysis and visualization.

**Funding:** The work of I.-F.P. is supported by Taiwan's Ministry of Science and Technology Grant MOST 107-2111-M-008-001-MY3. The work of J.C.L.C. is supported by the Research Grants Council of Hong Kong Grant E-CityU101/16. The work of I.-I.L. is supported by Taiwan's Ministry of Science and Technology (MOST 106-2111-M-002-011-MY3, MOST 108-2111-M-002-014-MY2). The work of K.T.F.C. is jointly supported by the National Natural Science Foundation of China (41775097), and the National Natural Science Foundation of China and Macau Science and Technology Development Joint Fund (NSFC-FDCT), China and Macau (41861164027).

**Acknowledgments:** The authors thank JTWC for Hato's track data, NASA for the MODIS data, NOAA for ETOPO2v2 datasets, SCSIO for SCSPOD14 climatology, ECMWF for the ERA-interim atmospheric reanalysis, and Macau Meteorological and Geophysical Bureau for the air temperature observations. Thanks also to the anonymous reviewers for their insightful and constructive comments that helped improve the quality of this work.

**Conflicts of Interest:** The authors declare no conflict of interest.

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
