# Peer review of "Rapid Intensification of Typhoon Hato (2017) over Shallow Water"

_sustainability, doi:10.3390/su11133709_

Round 1
Reviewer 1 Report
Review of “Sudden Intensification of Typhoon Hato (2017) over Shallow Water”
General comments:
This paper deals with the effect of warm shallow water on tropical cyclone intensification and shows a case of Typhoon Hato (2017) that experienced rapid intensification over a nearshore shallow water area in the South China Sea. This paper is interesting because it shows possibilities that warm, coastal shallow water, where ocean heat content is relatively low, plays an important role in Hato’s rapid intensification and the effect can be quantitatively evaluated through a diagnostic model. I have some concerns and questions about the present manuscript. But, once these issues are addressed, I believe this manuscript to be published.
Major comments:
1. In the diagnosis through the Price 2009 model, the authors estimated mixing depth by assuming the mixed layer depth (MLD) and vertical temperature gradient derived from climatology. Are the variability of the MLD and vertical temperature gradient, in general, low around the region of interest? Considering the fact that SST was abnormally high, isn’t there a possibility that the MLD was also deeper than usual? In fact, Fig. 8 shows that the MLDs at P2 and P3 points are much deeper than those of Fig. 4. Estimations of SST cooling depend on the assumption of the depth of the MLD. The authors used an ocean analysis dataset named MODAS in the LZSNFS simulation. Why didn’t the authors use the analysis dataset for the estimation of mixing depth by Price 2009 model?
2. I wonder if the term “sudden intensification” is appropriate for expressing Hato’s intensification just before the landfall. In fact, the intensification of Hato started when it was located in the Bashi Channel. It is not like Hato started rapid intensification just before the landfall. But, the use of “sudden” implies the onset of rapid intensification just before the landfall. I think the authors need to reconsider the wording.
3. The examination of atmospheric environmental conditions is not enough to conclude that the conditions were favorable for intensification. For example, Kaplan et al. (2010) used 8 predictors for their rapid intensification prediction. I think the authors cannot make a conclusion only with the information about vertical wind shear and humidity. Additionally, vertical wind shear of 9-11 m/s is, in general, unfavorable for intensification. For example, according to Fig. 3 of Rios-Berrios and Torn (2017), vertical wind shear of 11 m/s in the western North Pacific can be categorized as strong. Wang Y. et al. (2015) showed that “a TC has a better chance to intensify than to decay when the deep-layer shear is lower than 7–9 m/s”. Thus, generally speaking, the shear condition was unfavorable for Hato’s intensification. Nevertheless, Hato did intensify. The authors may take into account recent studies hypothesizing that other factors, for example, high SST, can help offset the negative effect of strong shear (e.g., Rios-Berrios&Torn, 2017).
Minor comments:
1. L67-70: This description is not correct. According to Figs. 1 and 2, the intensification of Hato started from 1800 UTC August 21 to 0300 UTC August 23. Thus, it is the latter half of the intensification that Hato was located over the shallow water area.
2. L99-102: the expression “20 kt per 6 hours” is incorrect and misleading. Correctly, 10 kt increased within 3 hours just before the landfall.
3. L141-143: How different is the atmospheric condition between the station observations and the ERA-interim data? Did this difference lead to differences in air-sea flux estimations?
4. L215: Is the MODIS SST used in this study skin SST or bulk SST (1-m depth SST)?
5. L222: I guess the actual MLD was thicker than climatological MLD. Is that right?
6. L335-337: Did the COAMPS simulation used as the surface forcing reproduce rapid intensification and maximum intensity? Is the resolution 0.2° correct? It seems too coarse to resolve intense tropical cyclones.
7. L338: What about the boundary condition?
8. L345: Why did the model simulate the ocean until the end of the year?
9. Table 6: At P2 point, the cooling was very different from that of Table 2. Why was that?
10. L368-369: Does “initial” ocean structure mean the ocean structure on May 1, 2017 (the initial time of the simulation) or the one just before Hato passed? What about the accuracy of the surface forcing? Which ocean structure was correct, that of Fig. 4 or that of the LZSNFS simulation?
11. L425-428: Did the authors check the actual SST cooling from in site observations? Fig. 13b shows the difference in SST between 1200 UTC August 21 and 1200 UTC August 23, but Fig. 7d shows the difference in SST between 0000 UTC August 21 and 0300 UTC August 23, which is the time when Hato just made landfall. The authors should look at a decrease in SST during the same period.
12. L452-454: Kaplan and DeMaria (2003) define the threshold of rapid intensification as the 95th percentile of intensity changes per day, not per 6 or 3 hours. Thus, comparing 10 kt per 3 hours of Hato’s intensification to the threshold of Kaplan and DeMaria (2003) is not suitable here.
13. L487: How about the effect of coastal upwelling?
Reviewer 2 Report
The paper discusses the rapid intensification of Typhoon Hato in 2017 at Macau. The air-sea combination models reveal that the shallow water warm system and its vertical thermal structure cause such intensification in sea bed like edge.
The observation data includes the high resolution satellite data. Several full ocean model are simulated to clear the heat moving and exchange system between sea and air. The both methods are reliable based on many review works.
The paper is suitable as a contribution to SUSTAINABILITY with miner revision.
1) Line 21; 20kt per to 6hours is included in “large intensification”? Please refer to the normal level of intensification.
2) Line 32: vertical mixing. >>> vertical mixing,
3) Line 115; SCSPOD14 can cover the sea surface temperature profiles over SSC? Is it coming from the satellite data? Please add the several additional experiments.
4) Table 1: Why 06:00 is indicate as 06Z ?
5) Line146: Main reason of Hato’s sudden intensification is the cooling of SCS temperature in shallow water ?
6) Figure 4: In the figure the temperature increases as the depth decreases. This is the simple relationship graph or not ? If not how do you obtain the relation >>
7) As conclusion, which direction dose the enthalpy f face vector to the sea surface or seaside?
Deference of results by the simulation models may be introduced in the introduction.
Round 2
Reviewer 1 Report
Review of “Sudden Intensification of Typhoon Hato (2017) over Shallow Water”
General comments:
The authors have addressed most of my concerns. I have one minor suggestion below.
Regarding rerunning of the Price 2009 model, I understand the author’s consideration. General readers, however, cannot understand the hidden consideration in the present form of the manuscript. I suggest the authors add it to the manuscript, i.e., why the authors used annual mean structures for the Price 2009 model, instead of monthly mean structures and MODAS analysis data.
